# RELGNN: Composite Message Passing for Relational Deep Learning

Tianlang Chen [1]   Charilaos Kanatsoulis [1]   Jure Leskovec [1]

## Abstract

Predictive tasks on relational databases are critical in real-world applications spanning e-commerce, healthcare, and social media. To address these tasks effectively, Relational Deep Learning (RDL) encodes relational data as graphs, enabling Graph Neural Networks (GNNs) to exploit relational structures for improved predictions. However, existing RDL methods often overlook the intrinsic structural properties of the graphs built from relational databases, leading to modeling inefficiencies, particularly in handling many-to-many relationships. Here we introduce RELGNN, a novel GNN framework specifically designed to leverage the unique structural characteristics of the graphs built from relational databases. At the core of our approach is the introduction of atomic routes, which are simple paths that enable direct single-hop interactions between the source and destination nodes. Building upon these atomic routes, RELGNN designs new composite message passing and graph attention mechanisms that reduce redundancy, highlight key signals, and enhance predictive accuracy. RELGNN is evaluated on 30 diverse real-world tasks from RELBENCH (Fey et al., 2024), and achieves state-of-the-art performance on the vast majority of tasks, with improvements of up to 25%.

## 1. Introduction

Predictive modeling over relational data (multiple tables connected via primary-foreign key relations) is central to numerous real-world applications: e-commerce platforms forecast product demand, music streaming services personalize recommendations, financial institutions assess credit risk, etc. The common strategy to tackle these predictive tasks relies on classical tabular machine learning approaches (Chen & Guestrin, 2016) that often require flattening relational data into a single table through manual feature engineering (Kaggle, 2022). This approach is not only labor-intensive but also leads to a substantial loss of predictive signal, as it oversimplifies the interconnected structure of relational data during the flattening process.

To overcome the limitations of tabular approaches, Fey et al. (2024) introduced *Relational Deep Learning (RDL)*, a new machine learning paradigm that enables end-to-end trainable neural networks to perform predictive modeling directly on relational databases. In RDL, relational data is represented as a *graph*, where each entity is represented as a node, and the primary-foreign key links between entities define the edges. This graph-based representation allows Graph Neural Networks (GNNs) (Gilmer et al., 2017; Hamilton et al., 2017) to serve as predictive models, capturing complex relational dependencies that traditional approaches overlook. Complementing this advancement, RELBENCH (Robinson et al., 2024) provides the first comprehensive benchmark for evaluating and developing RDL models.

Building effective RDL models is essential for tackling predictive tasks, yet remains quite challenging: *Relational entity graphs* are large-scale, heterogeneous, and temporally dynamic. The models introduced by Robinson et al. (2024) apply standard heterogeneous GNNs (Schlichtkrull et al., 2018; Hu et al., 2020) to relational entity graphs. This design choice can be suboptimal, since standard heterogeneous GNNs are developed for generic multi-relational graphs, where each edge-type denotes a direct semantic interaction between nodes. In relational databases, however, edges are defined by *primary–foreign key* links that merely record table connectivity, rather than a native semantic relation. Treating these schema-driven links as ordinary semantic edges overlooks the database's characteristic topology (e.g., many-to-many motifs, as discussed later), misrepresents how information should propagate, and ultimately limits model fidelity. These differences motivate the design of architectures tailored to the structural regularities of relational databases.

**Our contribution.** We analyze the distinctive structural patterns of *relational entity graphs*. Our work builds upon a central insight: Relational databases abound with *many-to-many relationships*; because each foreign key can reference

[1]Computer Science Department, Stanford University. Correspondence to: Tianlang Chen <tlchen@cs.stanford.edu>.

*Proceedings of the 42nd International Conference on Machine Learning*, Vancouver, Canada. PMLR 267, 2025. Copyright 2025 by the author(s).

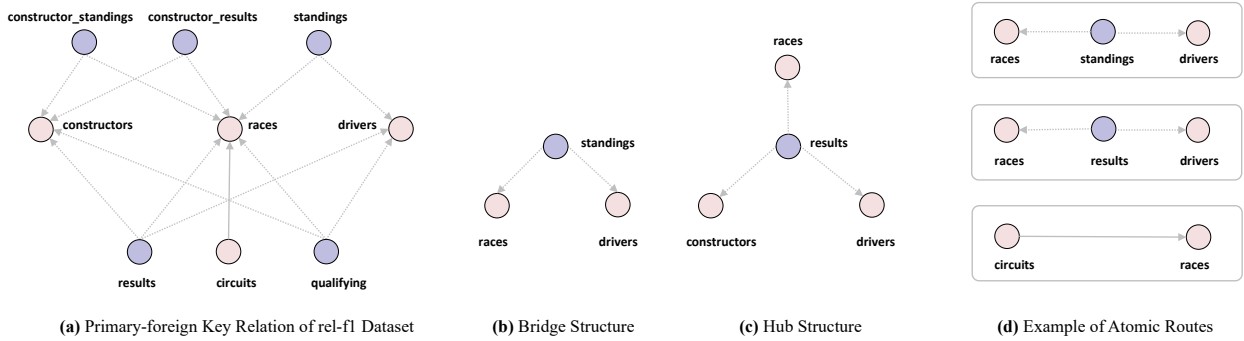

**(a)** Primary-foreign Key Relation of rel-f1 Dataset     **(b)** Bridge Structure     **(c)** Hub Structure     **(d)** Example of Atomic Routes

Figure 1: An illustration of the key concepts in our method. **(a)** The primary-foreign key relation of the `rel-f1` dataset. Arrows point from a node with a foreign key to the node with the corresponding primary key. Nodes with zero or one foreign key are marked in pink, and the corresponding foreign key is illustrated by a solid line. Nodes with two or more foreign keys are marked in purple, and the corresponding foreign keys are illustrated by dotted lines. **(b)** An example of the bridge structure, where *standings* node is a bridge node. **(c)** An example of the hub structure, where *results* node is a hub node. **(d)** Examples of three atomic routes, where the nodes within each box constitute a distinct atomic route.

only one primary key, these associations cannot be encoded by a single primary–foreign key edge. Instead, they are materialized through *junction* tables that sit between the two entity tables, decomposing each many-to-many association into a pair of one-to-many links. In the resulting graph, these tables give rise to two common substructures: (i) bridge nodes (cf. Figure 1(b)), which contain exactly two foreign keys and connect pairs of entities via tripartite pattern *(node-type 3 ← node-type 1 → node-type 2)* and (ii) hub nodes (cf. Figure 1(c)), which possess three or more foreign keys, forming star-shaped subgraphs. Bridge and hub nodes function mostly as 'routers', i.e., they contribute little semantic content, but channel information among their neighbors. Standard message passing GNNs repeatedly aggregate signals through the same bridge or hub nodes, generating over-smoothed representations. Moreover, the star-shaped motifs around hub nodes encode latent clique-like dependencies that conventional message passing cannot exploit.

To address these challenges, we propose RELGNN, a novel GNN framework that introduces the *composite message passing* and graph attention mechanism to fully exploit the unique structural properties of graphs built from relational database. RELGNN introduces the concept of *atomic routes*, which are simple paths that support complete, single-hop information exchange between the source and destination nodes, grounded in primary–foreign key relationships (cf. Figure 1(d)). When a table has a single foreign key, an atomic route consists of a pair of node-types and the edge connecting them. When a table has multiple foreign keys, it induces a set of atomic routes in the form of *(source → intermediate → destination)* paths, each of which connects two entities through a shared intermediate node (i.e., a bridge or hub node). Although reminiscent of meta-paths (Sun et al.,

2011) used in conventional heterogeneous graphs, atomic routes differ fundamentally in how they are constructed. While meta-paths typically rely on manual design guided by domain expertise, atomic routes are automatically and systematically derived from primary–foreign key relationships in relational databases, making them both broadly applicable and highly scalable across diverse datasets. Building on these routes, RELGNN designs composite message passing with attention mechanism that directly aggregates messages along atomic routes in a single step, avoiding redundant hops and preventing irrelevant information aggregation. This enables more efficient and accurate extraction of predictive signals compared to conventional heterogeneous GNNs.

We assess the performance of the proposed RELGNN across all tasks in RELBENCH (Fey et al., 2024), a benchmark which spans seven diverse relational databases covering e-commerce, social networks, sports, and medical platforms. RELBENCH features 30 real-world predictive tasks cast as entity classification, entity regression, and recommendation. RELGNN surpasses all baselines on 27 of the 30 tasks while performing comparably on the remaining three. Notably, RELGNN achieves more than a 4% improvement over a standard heterogeneous GNN in 17 out of 30 tasks, and provides up to a 25% improvement on the `site-success` regression task in the `rel-trial` database.

## 2. Preliminaries

### 2.1. Relational Database

A relational database $(\mathcal{T}, \mathcal{L})$ consists of a set of tables $\mathcal{T} = \{T_1, \ldots, T_n\}$ and a set of links between them $\mathcal{L} \subseteq \mathcal{T} \times \mathcal{T}$. Each table is a set $T = \{v_1, ..., v_{n_T}\}$, where the elements

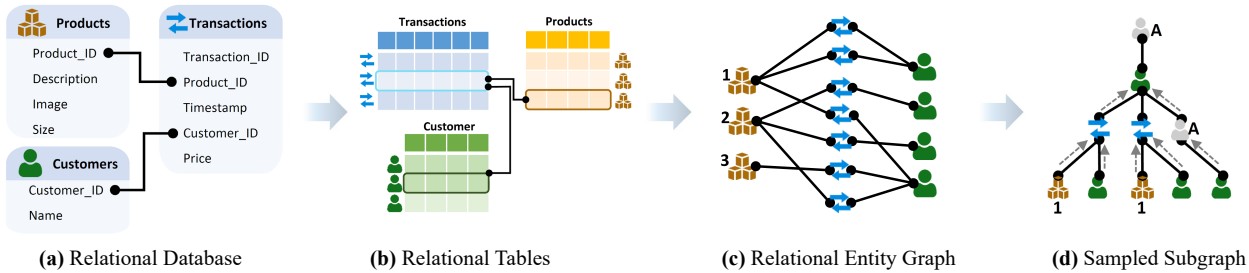

(a) Relational Database     (b) Relational Tables     (c) Relational Entity Graph     (d) Sampled Subgraph

Figure 2: Illustration of key concepts of RDL. **(a)** An example relational database. **(b)** Relational tables connected by primary-foreign key relations. **(c)** Relational entity graph built from relational tables. **(d)** Subgraph sampled with termporal neighbor sampling. Figures from Fey et al. (2024).

$v_i \in T$ are called rows or entities. Each entity $v \in T$ has a unique **primary key** $p_v$ that distinguishes it from other entities within the table. An entity can possess one or more **foreign keys**. Formally, each foreign key $\mathcal{K}_v \subseteq \{p_{v'} : v' \in T' \text{ and } (T, T') \in \mathcal{L}\}$ links an entity $v \in T$ to entity $v' \in T'$, where $p_{v'}$ is the primary key of $v'$ in table $T'$. An entity may have several attributes $x_v$, which represent the informational content, and may optionally include a timestamp $t_v$, indicating when the corresponding event occurred. Primary keys, foreign keys, attributes, and the timestamp are all realized as table columns. A link $L = (T_{\text{fkey}}, T_{\text{pkey}})$ between tables exists if a foreign key column in $T_{\text{fkey}}$ references a primary key column of $T_{\text{pkey}}$.

In Figure 2 (a), the TRANSACTIONS table contains one primary key column (TRANSACTIONID), two foreign key columns (PRODUCTID and CUSTOMERID), a numerical attribute column (PRICE), and a timestamp column (TIMESTAMP). The PRODUCTS table, by contrast, has a single primary key column (PRODUCTID), no foreign keys, and three attribute columns (DESCRIPTION, IMAGE, SIZE); it does not record timestamps. The black lines illustrate the foreign-key→primary-key links between the tables.

To capture the structural relationships between tables, we introduce the *schema graph*, which encodes the table-level topology of a relational database. Given a relational database $(\mathcal{T}, \mathcal{L})$, the schema graph is defined as $(\mathcal{T}, \mathcal{L})$, where each table (i.e., node-type) corresponds to a node, and each primary–foreign key relationship defines a directed edge from the foreign key table to the primary key table. Figure 1 (a) illustrates the schema graph of `rel-f1` dataset and Figure 4 shows the schema graphs of all the other datasets in RELBENCH. This definition corresponds to the directed variant of the schema graph described in Fey et al. (2024). Note that a node in the schema graph corresponds to a node-type in the relational entity graph, which we introduce in the next subsection.

## 2.2. Relational Deep Learning (RDL)

Fey et al. (2024) introduced RDL, an end-to-end framework for predictive modeling on relational databases with neural networks. RDL represents the database as a **relational entity graph**, which is a temporal, heterogeneous graph, where each table becomes a node-type, each row (entity) an individual node, and every primary–foreign key relationship an edge (Figure 2 (b)–(c)). The columns of each table provide the initial feature vector for their corresponding nodes. The transformation from the relational database to its relational entity graph form is lossless, preserving all information contained in the original tables.

Time is a first-class citizen in RDL. Each entity $v$ may carry a timestamp $t_v$ that records when the corresponding event occurred—for example, every row in the TRANSACTIONS table logs the moment of purchase. Many tasks are inherently temporal (e.g., predicting next-week product sales), so the model must respect causality. RDL therefore employs *temporal neighbor sampling* (Hamilton et al., 2017; Fey et al., 2024): for a seed entity at time $t$, it builds a subgraph that contains only nodes with timestamps $\leq t$, excluding all future information to avoid leakage. A GNN is then trained end-to-end on these time-consistent subgraphs, eliminating the need for manual feature engineering (Figure 2 (d)).

## 3. Bridge and Hub Topology in RDL (our work)

### 3.1. Structural Patterns in Relational Databases

**Heterogeneous graph topology.** General heterogeneous graphs encode information between nodes and edges of different types. The basic structural unit of the heterogeneous graph schema is a relation, written as a triplet (*source node-type, edge-type, destination node-type*). For example, a retail purchase is captured by the triplet *(customer, transaction, item)*, which represents a customer purchasing an item. Heterogeneous GNNs (Schlichtkrull et al., 2018;

Hu et al., 2020), are designed to process these relations through edge-semantic message passing, e.g., a message originating at a *customer node* traverses a *transaction edge* and is aggregated at the corresponding *item node* in a single hop. This message passing mechanism learns representations of both nodes and edge-types, and preserves the distinct semantics of each relation.

**Relational entity graph topology.** Unlike general heterogeneous graphs, where an edge denotes a semantically meaningful relation, the edges of relational entity graphs are formed by references to primary foreign keys in the database schema. Hence, each edge-type carries no intrinsic semantics beyond "table A points to table B," and the basic structural unit collapses to a pair *(node-type 1, node-type 2)*, rather than a semantic triplet. Consider the retail example: instead of a single relation *(customer, transaction, item)*, the database materializes a junction table *transaction*. In the relational entity graph, this becomes an explicit intermediate node-type, producing two edges *(transaction, customer)* and *(transaction, item)*. Because these edges encode only primary-foreign key relationships—not the semantic of purchasing—the usual message passing assumptions for heterogeneous graphs no longer hold. This shift from semantic triplets to purely structural links therefore calls for further investigation.

**Many-to-many relationships.** The transition from semantic triplets to primary–foreign key edges significantly impacts how many-to-many relationships are represented in relational databases. A many-to-many relationship describes a scenario where multiple instances of one entity type are associated with multiple instances of another. Although many-to-many relationships are ubiquitous, they cannot be directly encoded using a single primary–foreign key connection, as by definition, a foreign key can reference only one unique primary key. To address this, databases introduce junction tables that decompose many-to-many relationships into pairs of one-to-many links. In the retail example, customers and items form a many-to-many relationship—each customer can purchase multiple items, and each item can be purchased by multiple customers. This association is mediated by a transaction node, which serves as an intermediate connector in the relational entity graph. Each transaction node contains a pair of foreign keys, one referencing a customer node and the other referencing an item node, thereby decomposing the many-to-many association into two one-to-many links.

The previous observations lead to the following categorization of node-types in relational entity graphs: (i) node-types with zero or one foreign key, and (ii) node-types with two or more foreign keys. To illustrate them, consider the `rel-f1` schema, which tracks all-time Formula 1 racing data since 1950 (cf. Figure 1 (a)). In this schema, *constructors*, *races*, and *drivers* each have zero foreign keys, and *circuits* has one.

*Constructor_standings*, *constructor_results*, and *standings* each have two foreign keys, and *results* and *qualifying* each have three. Node-types with zero or one foreign key, highlighted in pink, do not require intermediate nodes to support message exchange and they exhibit a standard topology. In contrast, node-types with two or more foreign keys, highlighted in purple, serve as structural intermediaries in the schema. Given the ubiquity of many-to-many relationships in real-world interactions, cases involving (ii) are common in relational entity graphs and play a central role in shaping information flow. Next, we analyze the limitations of standard message passing over these structures and motivate a more specialized approach.

## 3.2. Challenges in Message Passing for RDL

### 3.2.1. NODE-TYPE WITH TWO FOREIGN KEYS (BRIDGE)

When a node-type has two foreign keys, it forms a subgraph of the form *(node-type 3 ← node-type 1 → node-type 2)*, creating a local tripartite structure among these three node-types. In this configuration, *node-type 1* simply acts as an aggregating bridge between *node-type 2* and *node-type 3*. Under standard heterogeneous GNNs, message passing over this structure involves two hops: from the source to the bridge, and then from the bridge to the destination. However, this two-hop communication introduces two inefficiencies. First, *redundancy*: information from the destination node is passed to the bridge in the first hop and then routed back to itself in the second, duplicating self-information. Second, *imbalance*: since the bridge node is an one-hop neighbor of the destination node while the source node is two hops away, the destination aggregates information from the bridge in both hops but receives information from the source only in the second. This leads to an overemphasis on the bridge node and under-utilization of the source node, which often carries more informative and predictive signals.

For example, consider the task of predicting race outcomes, where information flows from a *races* node (source) to a *drivers* node (destination) via a *standings* node (bridge) (cf. Figure 1 (b)). In standard two-hop message passing, information from *standings* reaches the *drivers* node in both hops, while information from *races* arrives only in the second. Since the *races* node typically contains more contextual information relevant to driver performance, this imbalance may degrade performance. As we show in the next section, modeling such structure as an atomic route enables direct one-hop message passing between the relevant node-types without any loss of information.

### 3.2.2. NODE-TYPE WITH THREE OR MORE FOREIGN KEYS (HUB)

When a node-type has three or more foreign keys, it forms a star-shaped subgraph that acts as a communication hub,

connecting multiple other node-types. For example, as shown in Figure 1 (c), the *results* node links *constructors*, *races*, and *drivers*, thereby mediating interactions among *constructors-races*, *races-drivers*, and *constructors-drivers*. These hub node-types inherit the same inefficiencies observed in the two-foreign key case (bridge structures): Standard two-hop message passing leads to redundant communication paths and imbalanced predictive signals since information from hub nodes is aggregated multiple times, while more informative signals from source nodes may be underrepresented. Furthermore, the star-shaped connectivity patterns around hub node-types induce latent second-order clique structures at the schema level—structures that are typically overlooked and under-utilized by standard modeling approaches. Our proposed approach, detailed next, is designed to exploit these latent clique structures, which constitute critical subgraphs in many real-world domains. This shift from star-shaped to clique-like connectivity substantially increases graph density and transforms the geometry of information flow.

## 4. Proposed Architecture: RELGNN

### 4.1. Atomic Routes in RDL

To address the limitations of standard message passing in relational entity graphs outlined above, we introduce the concept of *atomic routes*.

**Definition 4.1** (Atomic Route). An atomic route is a simple path between node-types that enables a single-hop inter-action between the source and destination node-type. We distinguish two cases:

1. **Single foreign key.** If a table has exactly one foreign key referencing to another table, the atomic route is an edge between the foreign key node-type (source) and the primary key node-type (destination), forming a simple path (*source → destination*).

2. **Multiple foreign keys.** If a table has multiple foreign keys referencing to different tables, the atomic routes are hyperedges that connect pairs of foreign key node-types (source and destination) via the intermediate primary key node-type, forming simple paths of type (*source → intermediate → destination*).

Figure 1 and 3 illustrate the primary-foreign key relationships in the rel-f1 dataset and the atomic routes derived from these relationships. For instance, the *circuits* table has only one foreign key, which points to the *races* table, forming atomic routes (*circuits → races*) and (*races → circuits*). In contrast, the *standings* table has two foreign keys connecting it to both the *drivers* and *races* tables. This results in atomic routes (*drivers → standings → races*) and (*races →*

*standings → drivers*). These routes capture the necessary interactions among multiple entities within a single step.

**Comparison to Meta-paths.** We compare atomic routes to meta-paths, a widely utilized concept in heterogeneous graph learning (cf. Section 6 for an extended discussion of meta-paths). Although both structures facilitate message propagation, they differ fundamentally in their construction and purpose. Meta-paths typically necessitate manual design guided by domain knowledge and task-specific intuition. This manual design process makes them labor-intensive, sensitive to dataset-specific characteristics, and susceptible to introducing selection bias. In contrast, atomic routes are derived automatically from the primary–foreign key relationships defined in the database schema. This enables their extraction without human supervision, ensuring applicability across a wide range of relational databases—including those with complex or previously unseen schemas. While meta-paths are often constructed to emphasize particular semantic patterns relevant to a task, atomic routes serve a fundamentally different function: they systematically capture the minimal pathways to avoid structural inefficiencies due to the primary-foreign key constraints.

### 4.2. Composite Message Passing for RDL

In this subsection, we build upon the concept of atomic routes and design composite message passing mechanisms for RDL. We begin this discussion by applying a standard heterogeneous GNN on a subgraph that encodes tables with multiple foreign keys. We assign src, dst, and mid to represent nodes corresponding to source, destination, and intermediate node-types, respectively. In standard heterogeneous GNNs, it takes two steps to complete the full information exchange. In the first step, each mid node aggregates information from all its neighbor nodes:

$$\mathbf{h}_{\texttt{mid}}^{(l+1)} = \text{UPD}(\{\{\mathbf{m}_R^{(l+1)}|\forall R = (\texttt{T}, \ \phi(\texttt{mid})) \in \mathcal{R}\}\}), \tag{1}$$

where

$$\mathbf{m}_R^{(l+1)} = \text{AGGR}(\mathbf{h}_{\texttt{mid}}^{(l)}, \{\{\mathbf{h}_u^{(l)}|\phi(u) = \texttt{T}\}\}),$$

$\mathbf{h}_v^{(l)}$ denotes the embedding of node $v$ at the $l$-th layer, UPD and AGGR are arbitrary differentiable functions with optimizable parameters, $\{\{\cdot\}\}$ denotes a permutation invariant set aggregator (e.g. mean, sum), $\mathcal{R}$ denotes the edge set consisting of pairs of node-types connected through primary-foreign key relationships and $\phi(\cdot)$ denotes a function mapping a node to its corresponding node-type. Then in the second step, the message passed from mid to dst is

$$\mathbf{m}_{(\texttt{mid, dst})}^{(l+2)} = \text{AGGR}(\mathbf{h}_{\texttt{dst}}^{(l+1)}, \{\{\mathbf{h}_{\texttt{mid}}^{(l+1)}\}\}) \tag{2}$$

Note that in Equation (1), T represents all node-types connected to the intermediate node-type. Therefore, in addition

to the information from the source node-type, information from other node-types connected to the intermediate node-type is also aggregated during this step. Furthermore, the information from `dst` is passed to `mid` in this step, and subsequently passed back to `dst` again in Equation (2), leading to redundancy, as discussed in Section 3.2.

To avoid these modeling inefficiencies we propose a novel composite message passing scheme based on atomic routes.

$$\mathbf{m}^{(l+1)}_{(\texttt{dst,mid,src})} = \text{AGGR}(\mathbf{h}^{(l)}_{\texttt{dst}}, \{\{\text{FUSE}(\mathbf{h}^{(l)}_{\texttt{mid}}, \mathbf{h}^{(l)}_{\texttt{src}})\}\})$$
$$(3)$$

Equation (3) describes a composite information exchange from `src` via `mid` to `dst` that is completed within a single step. As a result, there is no extraneous information entangled in the process. In summary, our approach effectively tackles the challenges that standard heterogeneous GNNs may encounter, such as multiple steps needed for complete information exchange and redundant aggregation during message passing.

### 4.3. RELGNN: Composite Message Passing with Atomic Routes

The introduction of atomic routes and composite message passing enables the design of new architectures specifically tailored to relational entity graphs. Equation (3) admits multiple instantiations, offering a flexible framework for message passing using atomic routes. Here, we propose RELGNN, a graph attention instantiation of Equation (3). When multiple foreign keys are involved, RELGNN instantiates Equation (3) as follows: for each `mid` node, it fuses information from each `src` node connected via a primary–foreign key relationship. Specifically, FUSE(·) is implemented as a linear combination:

$$\text{FUSE}(\mathbf{h}^{(l)}_{\texttt{mid}}, \mathbf{h}^{(l)}_{\texttt{src}}) = \mathbf{W}_1 \mathbf{h}^{(l)}_{\texttt{mid}} + \mathbf{W}_2 \mathbf{h}^{(l)}_{\texttt{src}}. \qquad (4)$$

Note that, due to the nature of foreign keys, each `mid` node is connected to only one `src` node. Then, RELGNN instantiates AGGR(·) with the standard multi-head attention mechanism (Vaswani et al., 2017; Shi et al., 2021), where embeddings from destination nodes serve as queries, and embeddings derived from the fusion operation in Equation (4) serve as keys and values. Let $\mathbf{h}^{(l)}_{\texttt{fuse}} := \text{FUSE}(\mathbf{h}^{(l)}_{\texttt{mid}}, \mathbf{h}^{(l)}_{\texttt{src}})$ as defined in Equation (4). AGGR(·) is realized as:

$$\text{AGGR}(\mathbf{h}^{(l)}_{\texttt{dst}}, \{\{\mathbf{h}^{(l)}_{\texttt{fuse}}\}\}) = \mathbf{W}_{\text{proj}} \mathbf{h}^{(l)}_{\texttt{dst}}$$
$$+ \sum_{\texttt{fuse} \in \mathcal{N}(\texttt{dst})} \alpha_{\texttt{dst,fuse}} \mathbf{W}_V \mathbf{h}^{(l)}_{\texttt{fuse}}, \qquad (5)$$

where the attention coefficients $\alpha_{\texttt{dst,fuse}}$ are computed via multi-head attention (with the multi-head notation omitted

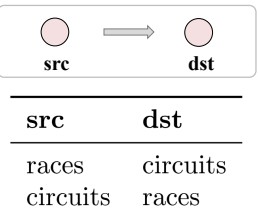

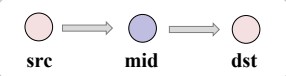

| src | dst |
|-----|-----|
| races | circuits |
| circuits | races |

| src | mid | dst |
|-----|-----|-----|
| races | constructor_standings | constructors |
| constructors | constructor_standings | races |
| races | constructor_results | constructors |
| constructors | constructor_results | races |
| drivers | standings | races |
| races | standings | drivers |
| drivers | results | races |
| races | results | drivers |
| constructors | results | races |
| races | results | constructors |
| drivers | results | constructors |
| constructors | results | drivers |
| drivers | qualifying | races |
| races | qualifying | drivers |
| constructors | qualifying | races |
| races | qualifying | constructors |
| drivers | qualifying | constructors |
| constructors | qualifying | drivers |

Figure 3: All the atomic routes derived from `rel-f1` dataset. The primary–foreign key relations of `rel-f1` is illustrated in Figure 1 (a).

for brevity):

$$\alpha_{\texttt{dst,fuse}} = \text{softmax}\left( \frac{(\mathbf{W}_Q \mathbf{h}^{(l)}_{\texttt{dst}})^\top (\mathbf{W}_K \mathbf{h}^{(l)}_{\texttt{fuse}})}{\sqrt{d}} \right).$$

In cases where tables with multiple foreign keys are not present, there is only a source and destination node-type, so the fusion operation is not needed. We directly substitute $\mathbf{h}_{\texttt{fuse}}$ with $\mathbf{h}_{\texttt{src}}$ in Equation (5):

$$\mathbf{m}^{(l+1)}_{(\texttt{dst,src})} = \text{AGGR}(\mathbf{h}^{(l)}_{\texttt{dst}}, \{\{\mathbf{h}^{(l)}_{\texttt{src}}\}\}) \qquad (6)$$

We use different weight matrices in Equation (4) and Equation (5) for each atomic route, enabling RELGNN to capture different types of information across routes.

Finally, the destination node aggregates information from all atomic routes related to it using a simple summation and

updates its embedding:

$$\mathbf{h}_{\text{dst}}^{(l+1)} = \sum_{t \in \mathcal{T}(\text{dst})} \mathbf{m}_t^{(l+1)}, \qquad (7)$$

where $\mathcal{T}(\text{dst})$ denotes the set of atomic routes with dst as the destination node, and $\mathbf{m}_t^{(l+1)}$ denotes the message from atomic route $t$, as defined in Equation (3) or Equation (6). Note that this final update operation will not lead to information entanglement, as the model learns distinct weight matrices for each atomic route, therefore learning to assign an appropriate weights to message from each atomic route during the summation.

# 5. Experiments

We evaluate RELGNN on RELBENCH (Robinson et al., 2024), a public benchmark designed for predictive tasks over relational databases using GNNs. RELBENCH offers a diverse collection of real-world relational databases and realistic predictive tasks. The benchmark covers 7 datasets, each carefully processed from real-world sources across diverse domains such as e-commerce, social networks, medical records, Q&A platforms, and sports. These datasets vary significantly in size, with differences in the number of rows, columns, and tables, serving as a challenging and comprehensive benchmark for RDL model evaluation. Appendix A.1 provides description and detailed statistics for each dataset.

RELBENCH introduces 30 predictive tasks covering a wide range of real-world use cases, grouped into three representative types: entity classification (Section 5.1), entity regression (Section 5.2), and recommendation (Section 5.3). These tasks are designed to reflect practical applications, such as predicting event attendance, estimating sales of an item, and recommending posts to users. The data is split temporally, with models trained on data from earlier time periods and tested on data from future time periods. To attach target labels, each task defines a *training table* that links entities of interest to their target labels and timestamps via foreign keys, enabling automatic supervision from historical data while ensuring temporal consistency during training. The tasks vary significantly in the number of entities in the train/validation/test split and the proportion of test entities encountered during training. Detailed description of each task can be found in Appendix A.2.

We follow the data processing pipeline introduced in Robinson et al. (2024). Relational data is transformed into heterogeneous temporal graphs, and temporal neighbor sampling is used to construct subgraphs centered around each entity and timestamp. Initial node embeddings are extracted from raw table attributes using PyTorch Frame (Hu et al., 2024). Final node embeddings are passed to prediction heads specific to the type of task to generate final predictions.

For baselines, we compare with the heterogeneous Graph-SAGE (Hamilton et al., 2017; Fey & Lenssen, 2019; Robinson et al., 2024) used in the original RELBENCH paper. To ensure a fair comparison, we maintain identical settings, including the data processing pipeline and training table, the temporal neighbor sampling algorithm, the initial node embeddings extraction model, the prediction head, and the loss function. We also incorporate a Light Gradient Boosting Machine (LightGBM) (Ke et al., 2017) as an additional non-RDL baseline, which is applied directly to the raw entity table features, following the setting in Robinson et al. (2024).

## 5.1. Entity Classification

**Experimental Setup.** The entity classification task involves predicting binary labels for a given entity at a specific seed time. The performance is evaluated with the ROC-AUC (Hanley & McNeil, 1983) metric, where higher values indicate better performance. The prediction head for this task consists of a multi-layer perceptron (MLP) applied to the GNN-generated node embeddings. The model is trained using binary cross-entropy loss. All results are averaged over five different seeds.

**Results.** Table 1 presents the results, along with the relative improvement of RELGNN over the standard heterogeneous GNN. RELGNN outperforms the baselines on 10 out of 12 tasks and achieves comparable performance on the remaining two. Notably, the relative gain is more significant on datasets with a more complex primary-foreign key structure (e.g., rel-f1; see Appendix B for visualizations). In contrast, performance improvements are smaller on datasets with simpler primary-foreign key structures, like rel-amazon and rel-hm. This trend aligns with the core design principle of RELGNN, demonstrating that its performance gains arise from effectively highlighting key predictive signals and eliminating redundant message aggregation—key challenges faced by standard heterogeneous GNNs when applied directly to relational entity graphs.

**Discussion and Limitations.** On the rel-stack dataset, RELGNN does not achieve significant improvements. A potential cause might be the unique self-loop structure, where primary-foreign key links connect nodes of the same type (posts). This pattern does not present in other datasets in RELBENCH and introduces an orthogonal modeling challenge. We provide a potential solution in Appendix D. Since robust handling of self-loops is orthogonal to our core contribution, we leave it for future work.

## 5.2. Entity Regression

**Experimental Setup.** Entity regression task requires predicting numerical labels for an entity at a specific seed time. The evaluation metric is Mean Absolute Error

Table 1: Entity classification results (ROC-AUC(%), higher is better) on RELBENCH test set. Best values are in bold.

| Dataset | Task | LightGBM | Hetero-GNN | RELGNN (ours) | Relative Gain |
|---|---|---|---|---|---|
| rel-amazon | user-churn | 52.22 | 70.42 | **70.99** | 1% |
| | item-churn | 62.54 | **82.81** | 82.64 | 0% |
| rel-avito | user-visits | 53.05 | **66.20** | 66.18 | 0% |
| | user-clicks | 53.60 | 65.90 | **68.23** | 4% |
| rel-event | user-repeat | 68.04 | 76.89 | **79.61** | 4% |
| | user-ignore | 79.93 | 81.62 | **86.18** | 6% |
| rel-f1 | driver-dnf | 68.56 | 72.62 | **75.29** | 4% |
| | driver-top3 | 73.92 | 75.54 | **85.69** | 13% |
| rel-hm | user-churn | 55.21 | 69.88 | **70.93** | 2% |
| rel-stack | user-engagement | 63.39 | 90.59 | **90.75** | 0% |
| | user-badge | 63.43 | 88.86 | **88.98** | 0% |
| rel-trial | study-outcome | 70.09 | 68.60 | **71.24** | 4% |

Table 2: Entity regression results (MAE, lower is better) on RELBENCH test set. Best values are in bold.

| Dataset | Task | LightGBM | Hetero-GNN | RELGNN (ours)) | Relative Gain |
|---|---|---|---|---|---|
| rel-amazon | user-ltv | 16.783 | 14.313 | **14.230** | 1% |
| | item-ltv | 60.569 | 50.053 | **48.767** | 3% |
| rel-avito | ad-ctr | 0.041 | 0.041 | **0.037** | 10% |
| rel-event | user-attendance | 0.264 | 0.258 | **0.238** | 8% |
| rel-f1 | driver-position | 4.170 | 4.022 | **3.798** | 6% |
| rel-hm | item-sales | 0.076 | 0.056 | **0.054** | 4% |
| rel-stack | post-votes | 0.068 | **0.065** | **0.065** | 0% |
| rel-trial | study-adverse | **44.011** | 44.473 | 44.461 | 0% |
| | site-success | 0.425 | 0.400 | **0.301** | 25% |

(MAE), where lower values indicate better performance. The prediction head is an MLP applied to GNN-generated node embeddings, identical to the setup used for entity classification. The model is trained using L1 loss. Results are averaged over five random seeds.

**Results.** Table 2 presents the results and the relative gain of our model over the standard heterogeneous GNN. REL-GNN outperforms the baselines on 8 out of 9 tasks and achieves comparable performance on the remaining one. As in the classification task, improvements are most pronounced on datasets with complex primary-foreign key structures, highlighting that RELGNN's effectiveness in modeling relational structure translates to consistent gains in both classification and regression settings.

**Discussion and Limitations.** The performance gain on rel-stack is limited, as discussed in Section 5.1. We also observe divergent results across tasks in the rel-trial dataset. A potential cause might be the inherent limitation of the prediction head for the entity regression task rather than the GNN model itself, as noted in the original RELBENCH paper (Robinson et al., 2024). In particular, study-adverse involves estimating an unbounded target (the number of patients with severe outcomes), while site-success predicts a bounded success rate. One possible explanation is that unbounded predictions may be more challenging given the inherently constrained design of the regression head. Enhancing prediction heads for regression tasks is an important direction for future work.

### 5.3. Recommendation

**Experimental Setup.** Recommendation tasks involve predicting a ranked list of top $K$ target entities for a given source entity at a specific seed time, where $K$ is predefined per task. This requires computing pairwise scores between source and target entities. We follow RELBENCH implementation and use two types of prediction heads: a two-tower

GNN (Wang et al., 2019a) and an identity-aware GNN (ID-GNN) (You et al., 2021). The two-tower GNN calculates pairwise scores through the inner product of source and target node embeddings and is trained using the Bayesian Personalized Ranking loss (Rendle et al., 2012). ID-GNN computes scores by applying an MLP to the embedding of the target entity within the subgraph sampled around each source entity and is trained using binary cross-entropy loss. Consistent with the original implementation, we use two-tower GNN for tasks in rel-amazon dataset and ID-GNN for tasks in the remaining datasets. The evaluation metric is Mean Average Precision (MAP) @$K$, where higher values indicate better performance. All results are averaged over five seeds.

**Results.** Results are presented in Table 3. REL-GNN achieves better or same performance compared to baselines on all 9 tasks. These consistent improvements across diverse datasets and tasks underscore the versatility of our approach and demonstrate its effectiveness in modeling complex relational information and extracting key predictive signals crucial for recommendation.

**Discussion and Limitations.** One limitation comes from the ID-GNN prediction head, which restricts recommendation candidates to nodes sampled within the source entity's subgraph for efficiency. This limitation is quantified by the *locality score* (Yuan et al., 2024), which measures the fraction of ground-truth targets present in the sampled subgraph. A high locality score suggests users tend to re-engage with previously interacted entities, while a low score indicates a preference for novel items. ID-GNN struggles on low-locality tasks where ground-truth targets often fall outside the sampled subgraph. Accordingly, RELGNN shows greater gains on high-locality tasks (e.g., rel-stack) than on low-locality ones (e.g., rel-hm, rel-trial). Designing improved prediction heads for recommendation tasks is orthogonal to our core contribution and remains a valuable direction for future work.

Table 3: Recommendation results (MAP(%), higher is better) on RELBENCH test set. Best values are in bold.

| Dataset | Task | LightGBM | Hetero-GNN | RELGNN (ours) | Relative Gain |
|---------|------|----------|------------|---------------|---------------|
| rel-amazon | user-item-purchase | 0.16 | 0.74 | **0.77** | 4% |
|  | user-item-rate | 0.17 | 0.87 | **0.92** | 6% |
|  | user-item-review | 0.09 | 0.47 | **0.52** | 11% |
| rel-avito | user-ad-visit | 0.06 | 3.66 | **3.94** | 8% |
| rel-hm | user-item-purchase | 0.38 | **2.81** | **2.81** | 0% |
| rel-stack | user-post-comment | 0.04 | 12.72 | **14.00** | 10% |
|  | post-post-related | 2.00 | 10.83 | **11.66** | 8% |
| rel-trial | condition-sponsor-run | 4.82 | 11.36 | **11.55** | 2% |
|  | site-sponsor-run | 8.40 | 19.00 | **19.14** | 1% |

## 6. Related Work

**Deep Learning on Relational Data.** Several works have explored the use of GNNs for learning on relational data (Schlichtkrull et al., 2018; Cvitkovic, 2019; Šír, 2021; Zahradník et al., 2023; Kanatsoulis et al., 2025). These works investigated different GNN architectures that utilize the relational structure. More recently, Fey et al. (2024) introduced Relational Deep Learning (RDL) (cf. Sec. 2.2), establishing a new subfield of machine learning. RDL has enabled various research opportunities, such as advancements in relational graph construction algorithms, GNN architectures, and task-specific prediction heads. Yuan et al. (2024) focused on improving recommendation tasks by addressing limitations of the currently employed two-tower and pair-wise prediction heads. In contrast, our work focuses on improving GNN architectures applied to all task types, offering an orthogonal contribution. Recently, a series of works have explored the use of large language models (LLMs) for heterogeneous graphs. HiGPT (Tang et al., 2024) introduced a language-enhanced heterogeneous graph tokenizer combined with LLMs. Wydmuch et al. (2024) proposed leveraging LLMs for predictive tasks in RDL, showing improvements on certain node-level tasks.

**Meta-paths in Heterogeneous Graphs.** Meta-paths (Sun et al., 2011) are widely used in heterogeneous graph learning to capture semantic relationships between different types of entities (Shang et al., 2016; Dong et al., 2017; Hu et al., 2018; Shi et al., 2018; Wang et al., 2019b; Fu et al., 2020). A meta-path is a manually designed sequence of node and edge-types in a heterogeneous graph intended to capture specific relational patterns. For example, in an academic graph, a meta-path *Author–Paper–Author* models co-authorship, and a meta-path *Author–Paper–Conference–Paper–Author* captures co-conference patterns. While effective in many settings, meta-paths have some known limitations (Shi et al., 2016; Hu et al., 2020; Shi, 2022). They typically rely on manual specification, which requires domain expertise and may introduce bias or overlook important patterns. Designing an effective set of meta-paths for complex graphs can be time-consuming and may fail to comprehensively capture all relevant interactions.

**Distinction between RDL and Knowledge Graphs.** The literature of knowledge graphs (Bordes et al., 2013; Wang et al., 2014; 2017) differs from RDL in terms of the tasks being tackled. Knowledge graph models mainly focus on *completion* tasks like predicting missing entities (e.g., Q: Who is the author of *Harry Potter*? A: J.K. Rowling) or missing relationships (e.g., Q: Did Yoshua Bengio win a Turing Award? A: Yes). In contrast, RDL focuses on making *predictions* about entities or groups of entities in the future timestamp (e.g., Will a customer churn in the next month? How much will a customer spend in the upcoming week?)

## 7. Conclusion

In this paper, we introduced RELGNN, a novel graph neural network framework specifically designed to address the structural inefficiencies of existing heterogeneous GNNs for relational databases. By leveraging atomic routes, we designed a composite message passing mechanism that enables direct single-hop interactions between the source and destination nodes. This avoids redundant aggregation and highlights key predictive signals, leading to more efficient and accurate predictive modeling. Through extensive evaluation on RELBENCH, a diverse benchmark covering 30 predictive tasks across 7 relational databases, RELGNN outperforms state-of-the-art baselines on the vast majority of tasks, achieving up to a 25% improvement in predictive accuracy. Our findings emphasize the limitations of conventional heterogeneous GNNs when applied to relational data and highlight the necessity of models that explicitly account for primary-foreign key relationships.

## Acknowledgments

We thank Rishabh Ranjan and Vijay Prakash Dwivedi for discussions and for providing feedback on our manuscript. We also gratefully acknowledge the support of NSF under Nos. OAC-1835598 (CINES), CCF-1918940 (Expeditions), DMS-2327709 (IHBEM), IIS-2403318 (III); Stanford Data Applications Initiative, Wu Tsai Neurosciences Institute, Stanford Institute for Human-Centered AI, Chan Zuckerberg Initiative, Amazon, Genentech, GSK, Hitachi, SAP, and UCB.

The content is solely the responsibility of the authors and does not necessarily represent the official views of the funding entities.

## Impact Statement

This paper presents work whose goal is to advance the field of Machine Learning. There are many potential societal consequences of our work, none which we feel must be specifically highlighted here.

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

Table 4: Statistics of RELBENCH.

| Dataset | Task name | Task type | #Rows of training table | | | #Unique Entities | %train/test Entity Overlap |
|---|---|---|---|---|---|---|---|
| | | | Train | Validation | Test | | |
| rel-amazon | user-churn | classification | 4,732,555 | 409,792 | 351,885 | 1,585,983 | 88.0 |
| | item-churn | classification | 2,559,264 | 177,689 | 166,842 | 416,352 | 93.1 |
| | user-ltv | regression | 4,732,555 | 409,792 | 351,885 | 1,585,983 | 88.0 |
| | item-ltv | regression | 2,707,679 | 166,978 | 178,334 | 427,537 | 93.5 |
| | user-item-purchase | recommendation | 5,112,803 | 351,876 | 393,985 | 1,632,909 | 87.4 |
| | user-item-rate | recommendation | 3,667,157 | 257,939 | 292,609 | 1,481,360 | 81.0 |
| | user-item-review | recommendation | 2,324,177 | 116,970 | 127,021 | 894,136 | 74.1 |
| rel-avito | user-clicks | classification | 59,454 | 21,183 | 47,996 | 66,449 | 45.3 |
| | user-visits | classification | 86,619 | 29,979 | 36,129 | 63,405 | 64.6 |
| | ad-ctr | regression | 5,100 | 1,766 | 1,816 | 4,997 | 59.8 |
| | user-ad-visit | recommendation | 86,616 | 29,979 | 36,129 | 63,402 | 64.6 |
| rel-event | user-repeat | classification | 3,842 | 268 | 246 | 1,514 | 11.5 |
| | user-ignore | classification | 19,239 | 4,185 | 4,010 | 9,799 | 21.1 |
| | user-attendance | regression | 19,261 | 2,014 | 2,006 | 9,694 | 14.6 |
| rel-f1 | driver-dnf | classification | 11,411 | 566 | 702 | 821 | 50.0 |
| | driver-top3 | classification | 1,353 | 588 | 726 | 134 | 50.0 |
| | driver-position | regression | 7,453 | 499 | 760 | 826 | 44.6 |
| rel-hm | user-churn | classification | 3,871,410 | 76,556 | 74,575 | 1,002,984 | 89.7 |
| | item-sales | regression | 5,488,184 | 105,542 | 105,542 | 105,542 | 100.0 |
| | user-item-purchase | recommendation | 3,878,451 | 74,575 | 67,144 | 1,004,046 | 89.2 |
| rel-stack | user-engagement | classification | 1,360,850 | 85,838 | 88,137 | 88,137 | 97.4 |
| | user-badge | classification | 3,386,276 | 247,398 | 255,360 | 255,360 | 96.9 |
| | post-votes | regression | 2,453,921 | 156,216 | 160,903 | 160,903 | 97.1 |
| | user-post-comment | recommendation | 21,239 | 825 | 758 | 11,453 | 59.9 |
| | post-post-related | recommendation | 5,855 | 226 | 258 | 5,924 | 8.5 |
| rel-trial | study-outcome | classification | 11,994 | 960 | 825 | 13,779 | 0.0 |
| | study-adverse | regression | 43,335 | 3,596 | 3,098 | 50,029 | 0.0 |
| | site-success | regression | 151,407 | 19,740 | 22,617 | 129,542 | 42.0 |
| | condition-sponsor-run | recommendation | 36,934 | 2,081 | 2,057 | 3,956 | 98.4 |
| | site-sponsor-run | recommendation | 669,310 | 37,003 | 27,428 | 445,513 | 48.3 |

# A. RELBENCH Details

In this section, we provide a detailed description and related statistics of RELBENCH. Table 4 provides detailed statistics for each dataset and task.

## A.1. Datasets

RELBENCH consists of 7 datasets, covering a diverse range of domains and scales. Below is a detailed description for each dataset.

**rel-amazon.** The Amazon E-commerce dataset contains product, user, and review interactions on Amazon's platform. It includes product metadata (e.g., price, category), review details (e.g., rating, text), and user engagement.

**rel-avito.** Avito, a major online marketplace, facilitates buying and selling across categories such as real estate, vehicles, and consumer goods. This dataset contains user search queries, ad characteristics, and additional contextual data for developing predictive models.

**rel-event.** The Event Recommendation dataset is derived from Hangtime, a mobile app that tracks users' social plans. It contains user interactions, event metadata, demographic information, and social network connections, offering insights into how social relationships influence user behavior.

**rel-f1.** The F1 dataset records comprehensive Formula 1 racing data since 1950, covering drivers, constructors, engine and tire manufacturers, and race circuits). It includes historical race results, season standings, and granular data on practice sessions, qualifying rounds, sprints, and pit stops.

**rel-hm.** The H&M dataset captures customer and product interactions from the retailer's e-commerce platform. It includes metadata on customers and products (e.g., demographic attributes, product descriptions), and purchase histories.

**rel-stack.** Stack Exchange is a network of Q&A websites where users earn reputation based on contributions. The dataset contains detailed activity logs, including user biographies, posts, comments, edit histories, votes, and linked questions.

**rel-trial.** The clinical trial dataset, sourced from the AACT initiative, aggregates study protocols and results. It includes trial design details, participant demographics, intervention specifics, and outcome measures, serving as a valuable resource for medical research and policy analysis.

## A.2. Tasks

The following list outlines the description predictive tasks included in RELBENCH.

1. rel-amazon

   (a) user-churn: Predict whether a user will stop reviewing products within the next three months.
   (b) item-churn: Predict whether a product will receive no reviews in the next three months.
   (c) user-ltv: Estimate the total dollar value of products a user will purchase and review over the next three months.
   (d) item-ltv: Estimate the total dollar value of purchases and reviews a product will receive in the next three months.
   (e) user-item-purchase: Predict the set of items a user will purchase in the next three months.
   (f) user-item-rate: Predict the set of items a user will purchase and rate five stars in the next three months.
   (g) user-item-review: Predict the set of distinct items a user will purchase and write a detailed review for in the next three months.

2. rel-avito

   (a) user-visits: Predict if a user will interact with multiple ads within next four days.
   (b) user-clicks: Predict if a user will engage with more than one ad by clicking within next four days.
   (c) ad-ctr: Estimate the click-through rate for an ad, assuming it receives a click within four days.
   (d) user-ad-visit: Predict the list of ads a user will visit within next four days.

3. rel-event

   (a) user-attendance: Predict the number of events a user will RSVP "yes" or "maybe" to in the next seven days.
   (b) user-repeat: Predict whether a user will attend an event (by responding "yes" or "maybe") in the next seven days, given they attended an event in the last 14 days.
   (c) user-ignore: Predict whether a user will ignore more than two event invitations in the next seven days.

4. rel-f1

   Node-level tasks:

   (a) driver-dnf: Predict whether a driver will fail to finish a race within the next month.
   (b) driver-top3: Predict if a driver will secure a top-three qualifying position in a race within the next month.
   (c) driver-position: Predict a driver's average finishing placement across all races in the next two months.

5. rel-hm

   Node-level tasks:

   (a) user-churn: Predict if a customer will become inactive (no transactions) in the next week.
   (b) item-sales: Predict total revenue generated by an article in the upcoming week.
   (c) user-item-purchase: Predict the list of articles a customer will over the next seven days.

6. rel-stack

(a) `user-engagement`: Predict whether a user will participate by voting, posting, or commenting within the next three months.

(b) `user-badge`: Predict if a user will earn a new badge within the next three months.

(c) `post-votes`: Predict the number of votes a user's post will receive over the next three months.

(d) `user-post-comment`: Predict which existing posts a user will comment on in the next two years.

(e) `post-post-related`: Identify a list of existing posts that will be linked to a given post within the next two years.

7. `rel-trial`

(a) `study-outcome`: Predict if a clinical trial will meet its primary outcome within the next year.

(b) `study-adverse`: Estimate the number of patients who will experience severe adverse events or death in a clinical trial over the next year.

(c) `site-success`: Predict the success rate of a trial site in the next year.

(d) `condition-sponsor-run`: Predict which sponsors will be associated with a particular condition.

(e) `site-sponsor-run`: Predict whether a specific sponsor will conduct a trial at a given facility.

## B. Visualization of Primary-Foreign Key Relationships

We visualize the primary-foreign key relationships of all datasets in RELBENCH in Figure 4 (`rel-f1` is visualized in Figure 1 (a)).

## C. Ablation Study

To isolate the contribution of atomic routes, we conduct an ablation study comparing variants of RELGNN with and without the attention mechanism. In our instantiation and main experiments, the FUSE operation of RELGNN is instantiated using the same design as GraphSAGE (cf. Equation (4)). For this ablation, we additionally configure the AGGR operation to use GraphSAGE-style aggregation instead of attention. This ensures that the only architectural difference between the baseline and RELGNN is the inclusion of atomic routes.

As shown in Tables 5, 6, and 7, RELGNN achieves consistent gains over the baselines, even when using the same aggregation mechanism. Notably, the performance improvement is not attributable to attention alone: the gap between RELGNN with and without attention is modest. These results indicate that the performance gains stem primarily from the use of atomic routes, which enable more efficient and targeted message passing across relational structures.

We also include additional baselines using GAT (Veličković et al., 2018) and GIN (Xu et al., 2019) as backbone architectures. As shown in Tables 5, 6, and 7, RELGNNoutperforms all baselines regardless of the backbone GNN.

## D. Additional Discussion

**Scalability and Efficiency in Large Relational Databases.** A common concern is whether the complexity of computing atomic routes increases with database size. In our framework, atomic routes are computed at the schema graph level, where nodes represent table types rather than individual entities. Even in large databases with millions of rows, the number of table types typically remains in the tens. This schema-level design makes RELGNN agnostic to the number of entities and allows it to scale efficiently as the database grows.

Another related concern is the complexity introduced by tables with many foreign keys. In practice, we follow the standard RDL sampling strategy, which samples a fixed-size local neighborhood around each seed node. As a result, only a subset of the foreign key links is activated at training or inference time. The number of active connections remains bounded by the sampling configuration, ensuring that the message passing remains efficient even when some tables connect to many others.

**A Potential Approach for Handling Self-Loop Tables.** Although self-loop tables are rare in relational databases (only 1 out of 7 datasets in RELBENCH), it is worthwhile to improve the model handling of such cases. The key to addressing this challenge is enhancing the model's ability to distinguish messages coming from the same table type (for self-loop tables) versus those from different types (for non-self-loop tables). To address this, we propose incorporating relative positional

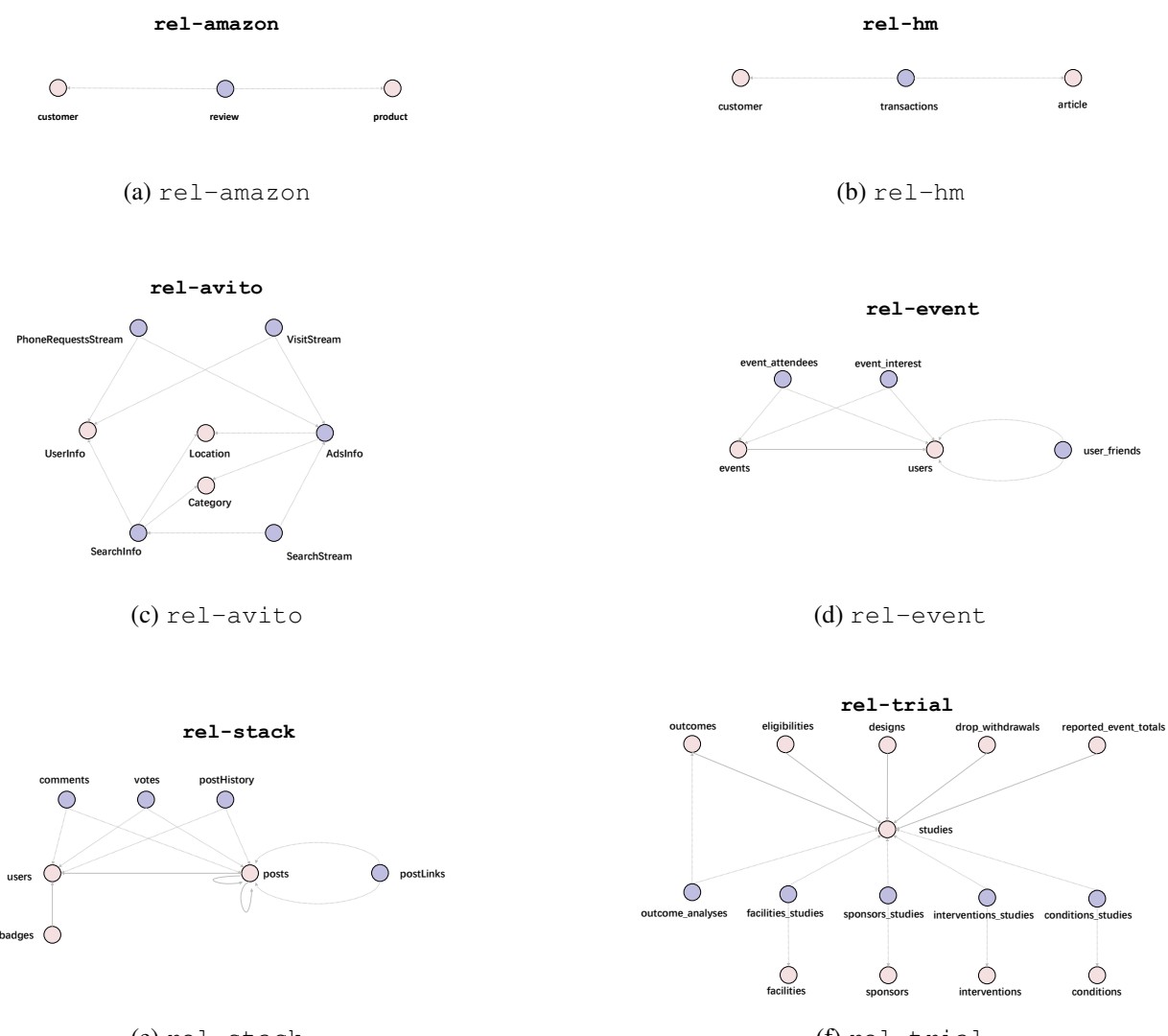

Figure 4: Primary-foreign key relationships of datasets in RELBENCH.

Table 5: Ablation results for entity classification (ROC-AUC(%), higher is better) on RELBENCH test set. The best values are in **bold**, and the second-best values are underlined.

| Dataset | Task | GraphSAGE | GAT | GIN | RELGNN | RELGNN w/o attn |
|---|---|---|---|---|---|---|
| rel-amazon | user-churn | 70.42 | 63.21 | 70.50 | **70.99** | 70.90 |
| | item-churn | 82.81 | 69.99 | 82.74 | 82.64 | **82.94** |
| rel-avito | user-visits | 66.20 | 64.82 | 65.96 | 66.18 | **66.80** |
| | user-clicks | 65.90 | 65.85 | 66.04 | **68.23** | 66.72 |
| rel-event | user-repeat | 76.89 | 68.24 | 74.35 | **79.61** | 78.82 |
| | user-ignore | 81.62 | 82.04 | 79.54 | **86.18** | 85.58 |
| rel-f1 | driver-dnf | 72.62 | 70.26 | 71.81 | **75.29** | 74.77 |
| | driver-top3 | 75.54 | 60.03 | 73.64 | **85.69** | 84.86 |
| rel-hm | user-churn | 69.88 | 64.72 | 69.91 | **70.93** | 70.29 |
| rel-stack | user-engagement | 90.59 | 89.59 | 90.53 | **90.75** | 90.70 |
| | user-badge | 88.86 | 84.51 | 88.72 | 88.98 | **88.99** |
| rel-trial | study-outcome | 68.60 | 66.19 | 68.44 | **71.24** | 69.34 |

Table 6: Ablation results for entity regression results (MAE, lower is better) on RELBENCH test set. The best values are in **bold**, and the second-best values are underlined.

| Dataset | Task | GraphSAGE | GAT | GIN | RELGNN | RELGNN w/o attn |
|---|---|---|---|---|---|---|
| rel-amazon | user-ltv | 14.313 | 16.626 | 14.318 | **14.230** | 14.240 |
| | item-ltv | 50.053 | 58.902 | 50.087 | 48.767 | **48.282** |
| rel-avito | ad-ctr | 0.041 | 0.043 | 0.041 | **0.037** | **0.037** |
| rel-event | user-attendance | 0.258 | 0.263 | 0.264 | **0.238** | **0.238** |
| rel-f1 | driver-position | 4.022 | 4.268 | 4.072 | 3.798 | **3.792** |
| rel-hm | item-sales | 0.056 | 0.079 | 0.055 | 0.054 | **0.053** |
| rel-stack | post-votes | **0.065** | 0.068 | **0.065** | **0.065** | **0.065** |
| rel-trial | study-adverse | 44.473 | 46.026 | **44.400** | 44.461 | 45.531 |
| | site-success | 0.400 | 0.393 | 0.398 | **0.301** | 0.354 |

Table 7: Ablation results for recommendation results (MAP(%), higher is better) on RELBENCH test set. The best values are in **bold**, and the second-best values are underlined.

| Dataset | Task | GraphSAGE | GAT | GIN | RELGNN | RELGNN w/o attn |
|---|---|---|---|---|---|---|
| rel-amazon | user-item-purchase | 0.74 | 0.44 | 0.69 | 0.77 | **0.90** |
| | user-item-rate | 0.87 | 0.78 | 0.78 | **0.92** | 0.89 |
| | user-item-review | 0.47 | 0.29 | 0.42 | 0.52 | **0.59** |
| rel-avito | user-ad-visit | 3.66 | 1.99 | 3.66 | 3.94 | **3.96** |
| rel-hm | user-item-purchase | 2.81 | 1.88 | 2.80 | 2.81 | **2.83** |
| rel-stack | user-post-comment | 12.72 | 11.97 | 12.81 | **14.00** | 13.66 |
| | post-post-related | 10.83 | 10.71 | 10.78 | **11.66** | 11.38 |
| rel-trial | condition-sponsor-run | 11.36 | 10.43 | 11.32 | **11.55** | 11.44 |
| | site-sponsor-run | 19.00 | 17.90 | 18.91 | **19.14** | 19.08 |

encoding (RPE) over the schema graph (where nodes are table types and edges are primary-foreign key relations). RPE helps the model identify whether two tables are of the same type and also offers additional benefits. Since the schema graph defines the database structure at a macro level, it can provide global relational context that complements the local structural information captured by GNNs on the relational entity graph. Moreover, schema graphs are typically small (with only tens of nodes even in large databases) and static, so the RPE can be precomputed once with negligible overhead and reused throughout training and inference.

To evaluate, we implemented an RPE method based on Huang et al.: eigen-decompose the Laplacian of the schema graph $L = V\text{diag}(\lambda)V^{\top}$, apply $m$ element-wise MLPs $\phi_k(\cdot)$ to obtain $Q[:, :, k] = V\text{diag}(\phi_k(\lambda))V^{\top}$, for $k \in [m]$, resulting in a tensor $Q \in \mathbb{R}^{N \times N \times m}$, and project via an MLP $\rho$ to obtain $RPE = \rho(Q) \in \mathbb{R}^{N \times N \times d}$, where $N$ is the number of node-types of $d$ is the dimension of node embeddings. For a message from node-type $i$ to $j$, we update the original message $m$ as $m' = m + \alpha \cdot RPE[i, j]$ with learnable $\alpha$. This resulted in a 2% improvement on both classification tasks in rel-stack, which contains self-loops.

**Integration with Advanced Components.** We focus on improving the core design of message passing, but the REL-GNN framework is flexible and accommodates more advanced components.

- **Advanced GNN Aggregators.** RELGNN allows easy integration of different GNN aggregators by re-instantiating the AGGR operation in Equation (5). For example, PNA (Corso et al., 2020) improves expressiveness by combining multiple aggregators and degree-scalers. It can be integrated into RELGNN by instatiating Equation (5) with $AGGR(\mathbf{h}_{dst}^{(l)}, \{\{\mathbf{h}_{fuse}^{(l)}\}\}) = \mathbf{W}_{\text{proj}}\mathbf{h}_{dst}^{(l)} + \sum_{fuse \in \mathcal{N}(dst)} M(\mathbf{h}_{fuse}^{(l)})$ where $M(\cdot)$ is the PNA operator. Model-agnostic aggregators can also be integrated on top of RELGNN. For instance, Jumping Knowledge Network (Xu et al., 2018) improves performance by adaptively combining representations from multiple layers. This can be implemented by collecting outputs from each RELGNN layer and passing them to PyG's JumpingKnowledge module, with the result used as input to the final prediction layer.

- **Effective Time Encodings.** Many RDL tasks involve temporal prediction, and various temporal encodings can be incorporated as additional node features for RELGNN. For example, Time2Vec (Kazemi et al., 2019) maps timestamps $t$ to $\mathbb{R}^d$ via $\text{Time2Vec}(t)[i] = sin(\omega_i t + \phi_i)$ with learnable parameters $\omega_i$ and $\phi_i$. GraphMixer (Cong et al.) introduces a fixed time encoding $t \to \cos(t\omega)$, where $\omega = \{\alpha^{-\frac{i-1}{\beta}}\}_{i=1}^d$ and the choice of $\alpha, \beta$ depends on the dataset's time range. These techniques are compatible with RELGNNand introduce minimal overhead.

