# OpenReview forum: "RelGNN: Composite Message Passing for Relational Deep Learning"
_ICML.cc/2025/Conference — ICML 2025 poster_

### Official Review · Reviewer_xbNa · 2025-02-27

**Overall Recommendation:** 3

**Summary:**

The paper proposes a graph neural network with attention mechanism, called RelGNN, for predictive tasks on relational tables. The paper introduces atomic routes based on primary-foreign key connections and design a composite message passing using atomic Routes. RelGNN achieves good results on RELBENCH that is a widely accepted benchmark for deep learning on relational data.

## update after rebuttal
My questions are answered. I maintain my original score.

**Claims And Evidence:**

Yes.

**Essential References Not Discussed:**

No

**Experimental Designs Or Analyses:**

Yes, I did. Experimental evaluations in the paper are conducted on a state-of-the-art benchmark.

**Methods And Evaluation Criteria:**

Yes, the proposed methods is evaluated on RELBENCH that is a widely accepted benchmark for deep learning on relational data.

**Other Comments Or Suggestions:**

Personal comment: I can not learn some implementation details as I did not find the source codes.

**Other Strengths And Weaknesses:**

Strengths
S1: Authors propose a simple but efficient solution for learning over relational tables.
S2: RelGNN have good performances on all tasks.
S3: The paper reads well.

Weakness
W1: Limitations of RelGNN are not discussed
W2: Implementations are not open-sourced.

**Questions For Authors:**

Q1: Will RelGNN degenerate on relational tables with many-to-many relationships? For example,  for movie and actor tables, one actors can appear in multiple movies and one movie can feature many actors.

Q2: The claim made in line 233 - 234 "This ensures both broad applicability and scalability across diverse datasets". Can authors explain what "broad applicability and scalability" means?

**Relation To Broader Scientific Literature:**

The key contributions of the paper are atomic routes and a composite message passing with atomic routes, which are simple but efficient. In general, simple but efficient approaches are easier to be deployed into systems.

**Theoretical Claims:**

Yes. I did read and understand the equations in the paper.

---

> ### Author Rebuttal · Authors · 2025-04-01
>
> We thank the reviewer for acknowledging RelGNN’s efficiency, strong performance, and the clarity of our writing. We also appreciate the constructive feedback and address each point below:
>
> > **Regarding Discussion on the Limitations (W1)**
>
> The reviewer mentions that the limitations of RelGNN are not discussed. In fact, we did discuss specific limitations within the experiment sections (e.g., discussing limitations on recommendation tasks due to the constraints of the inherited framework and prediction head from RDL; analyzing potential reasons for performance variations across tasks). We recognize these discussions and insights could be more accessible. In the revised version, we will add a consolidated limitations section. This reorganization will improve clarity without requiring substantial new content, as the critical analysis is already present in our original manuscript.
>
> > **Regarding Open-Sourcing Implementations (W2)**
>
> This is a good point and we acknowledge the importance of reproducibility and transparency. We've now made our source code available at  https://anonymous.4open.science/r/RelGNN.
>
> > **Regarding Handling Many-to-Many Relationships (Q1)**
>
> The reviewer inquires about RelGNN’s ability to handle many-to-many relationships. The answer is an unambiguous yes. Many-to-many relationships are common in relational databases and are routinely modeled using bridge tables. For instance, in the rel-hm dataset, a customer may purchase multiple articles, and an article may be purchased by many customers; similarly, in the rel-f1 dataset, a driver participates in multiple races while each race includes multiple drivers. Such relationships are handled through bridge tables (e.g., transaction tables in rel-hm dataset) which is standard practice in relational databases. Incorporating these bridge tables are necessary by definition of database - as primary keys must uniquely identify rows, many-to-many relationships require intermediate tables with foreign keys pointing to both related entities. In the reviewer's movie-actor example, a valid relational database design requires more than just "actors" and "movies" tables. To represent this many-to-many relationship correctly, a bridge table (commonly called "appearances" or "cast") must be introduced. This bridge table would contain foreign keys referencing the primary keys of both the "actors" and "movies" tables, thereby preserving referential integrity while enabling the many-to-many relationship.
>
> In fact, this prevalence of bridge-node-modeled many-to-many relationships distinguishes relational data graphs from general heterogeneous graphs, which motivated RelGNN's design to specifically leverage this unique structure, leading to improved performance compared to general heterogeneous GNNs. We thank the reviewer for this great question and will clarify this point in the revised manuscript.
>
> > **Regarding Clarification on “Broad Applicability and Scalability”**
>
> "Broad applicability and scalability" refers to two key advantages:
>
> - Broad Applicability: RelGNN's atomic routes can be automatically extracted without domain-specific knowledge (unlike traditional meta-path approaches), making it applicable to diverse relational datasets beyond Relbench.
> - Scalability: Atomic routes are computed at the schema graph level—a graph where nodes represent table types (which typically number in the tens even for large databases) rather than the data graph level, where nodes represent individual entities (which can number in the millions). This design makes our method agnostic to the number of entities, allowing RelGNN to scale efficiently to very large databases where the number of entities grow rapidly while table types remain relatively stable.
>
> We thank the reviewer for bringing this up and will clarify this explanation in the revised manuscript.
>
> In summary, we
>
> - Note that limitations were discussed in our experiment section and commit to a more explicit discussion in the revised version.
> - Make our source code available.
> - Clarify that many-to-many relationships are handled via bridge nodes, a necessity in relational database design.
> - Explain "broad applicability and scalability" by highlighting RelGNN’s independence from human knowledge and its efficiency as the database scales.
>
> We hope these clarifications address the reviewer’s concerns and will update the manuscript when permitted.

---

> > ### Comment · Reviewer_xbNa · 2025-04-02
> >
> > Thanks. I have no further questions.

---

> > > ### Author Response · Authors · 2025-04-04
> > >
> > > We thank the reviewer for the prompt response and are glad to hear that all questions have been addressed.
> > >
> > > We sincerely appreciate the reviewer’s thoughtful feedback in helping us improve the quality of our paper. If the reviewer finds it appropriate, we would be grateful for a re-evaluation and a potential update to the rating in light of our response.

---

### Official Review · Reviewer_935g · 2025-03-04

**Overall Recommendation:** 3

**Summary:**

The paper introduces RelGNN, a graph neural network (GNN) framework specifically designed for Relational Deep Learning (RDL), the task of doing end-to-end predictive modeling directly on relational databases (multiple tables linked by primary/foreign keys). RelGNN leverages  “atomic routes,” which reflect short tripartite or star-shaped connectivity among tables that have multiple foreign keys. These are sequences (or hyperedges) of node-types that facilitate more direct single-hop message passing between relational tables that are semantically connected.

## update after rebuttal
My concerns are addressed. Considering that I gave a relatively high score in the review, I will just maintain my score.

**Claims And Evidence:**

Claim 1: RelGNN better models relational DB structure through “atomic routes” than standard heterogeneous GNNs.

Evidence of Claim 1: The authors highlight the difference between typical “meta-path” approaches in heterogeneous graphs and the new approach of extracting routes directly from primary–foreign key constraints. The authors then show strong empirical improvements (≥15/30 tasks with >4% relative gain) on the RELBENCH benchmark.

Claim 2: Gains are larger on DBs with more complicated foreign-key structures (“bridge” or “hub” shapes).

Evidence of Claim 2: They measure the improvements by listing results across all tasks and highlight bigger improvements in “complex” schemas (e.g. rel-f1).

**Essential References Not Discussed:**

Given the scope, they mostly reference the standard GNN and relational DB learning papers. Possibly referencing more about “multi-hop GNN minimization / skipping aggregator nodes” from other design patterns might help, but not necessarily “essential.” No glaring missing references stand out.

**Ethical Review Concerns:**

N.A.

**Experimental Designs Or Analyses:**

They follow the standard approach from RELBENCH. Each of the 30 tasks is tested with a consistent data split (temporal). Baselines are run with the same node embedding initialization and the same sampling method. Besides, the authors highlight that improvements are especially large when the relational schema is complicated. That matches the paper’s central premise. The experiments appear valid for the stated goal. There do not appear to be unusual or confounding design choices in their methodology.

**Methods And Evaluation Criteria:**

Overall, the proposed methods and metrics make sense for the targeted RDL tasks.

**Other Comments Or Suggestions:**

1. A small ablation might be helpful: e.g. how does the removal of the composite step or the removal of atomic routes degrade performance?
2. The authors could incorporate or discuss how “atomic routes” would handle self-referencing foreign keys or cyclical references.
3. Please check the formating of the tables and the margins on the last page of the paper, as they are poorly presented.

**Other Strengths And Weaknesses:**

Stengths:
- Novel approach to dealing with multi-foreign-key bridging nodes, which is common in real relational DBs.
- Relatively strong results on a large set of real tasks from RELBENCH.
- Minimal overhead or domain knowledge needed: the “atomic route” concept is systematically derived from foreign key constraints.

Weaknesses:
- May not handle self-joins or self-loop tables elegantly (they mention rel-stack’s performance is not improved as much).
- The proposed method focuses on composite message passing but does not discuss integration with more advanced GNN aggregator designs or advanced time encodings.

**Questions For Authors:**

1. For tasks like rel-stack with self-joins (post to post), do you see a better representation approach than treating them like normal foreign-key edges?
2. For extremely large DBs, do atomic routes lead to any complexities in memory usage or graph construction time?
3. For tables with many foreign keys, do you risk a large blow-up in “composite messages” to handle all pairwise or triple connections?

**Relation To Broader Scientific Literature:**

The authors situate their approach relative to Relational Deep Learning (Fey et al., 2024) and the concept of knowledge graphs in heterogeneous GNNs. The difference is that knowledge graphs revolve around semantic relation types (like “author-of,” “works-in,” etc.), whereas relational DB edges come strictly from primary–foreign key constraints.
They also place it in context with meta-path approaches for heterogeneous GNNs (like HAN, R-GCN). However, they argue that meta-paths require domain knowledge, whereas they systematically build “atomic routes” from DB schema keys.
To conclude, they do address the main relevant lines of prior work on heterogeneous graph modeling and knowledge graph completions.

**Theoretical Claims:**

The paper does not heavily focus on new theoretical proofs. The main conceptual claim is that “atomic routes” in relational data can be grouped for single-step message passing. The authors do not provide formal theorems, but they do discuss the correctness of capturing needed information in one pass.

---

> ### Author Rebuttal · Authors · 2025-04-01
>
> We thank the reviewer for the constructive comments. We appreciate the recognition of our method’s novelty, strong results, and the clarity of our experimental setup. We address each point below:
>
> > **Regarding Handling Self-Loop Tables (W1, C2, Q1)**
>
> The reviewer suggests discussing solutions for self-loop tables, which we previously identified as a potential limitation. Although self-loop tables are rare in relational databases—only 1 out of 7 datasets in Relbench—we agree that discussing potential solutions is important. We experimented with adding positional encoding to help the model distinguish messages from different types of tables, which resulted in a 2% improvement on both entity classification tasks in the rel‑stack dataset. We will include a discussion in the revised version.
>
> > **Regarding Discussion of Integrating Other Components (W2)**
>
> The reviewer suggests discussing advanced GNN aggregators and time encodings. We find this an interesting point and will include a discussion on how our framework can be flexibly integrated with these components. We note that incorporating such elements is complementary and orthogonal to our primary focus on enhancing the message passing design for RDL. To ensure a fair comparison, we kept other components the same as prior work.
>
> > **Regarding Ablation Study (C1)**
>
> We appreciate the suggestion and incorporate an ablation study to further examine the impact of atomic routes. We instantiated Eq5 of RelGNN with GraphSAGE, which is equivalent to GraphSAGE + atomic routes. Additionally, we removed atomic routes from RelGNN, reducing it to a heterogeneous GAT. As shown in Table 1-3, the performance gap in both cases (GraphSAGE vs RelGNN w/ GraphSAGE; GAT vs RelGNN) highlights the effectiveness of atomic routes. We will include this ablation study in the revised version.
>
> **Table 1. Classification (ROC-AUC, ↑)**
> |Task|GraphSAGE|RelGNN w/ GraphSAGE|GAT|RelGNN|
> |-|-|-|-|-|
> |user-churn|70.42|70.90|63.21|70.99|
> |item-churn|82.81|82.94|69.99|82.64|
> |user-visits|66.20|66.80|64.82|66.18|
> |user-clicks|65.90|66.72|65.85|68.23|
> |user-repeat|76.89|78.82|68.24|79.61|
> |user-ignore|81.62|85.58|82.04|86.18|
> |driver-dnf|72.62|74.77|70.26|75.29|
> |driver-top3|75.54|84.86|60.03|85.69|
> |user-churn|69.88|70.29|64.72|70.93|
> |user-engagement|90.59|90.70|89.59|90.75|
> |user-badge|88.86|88.99|84.51|88.98|
> |study-outcome|68.60|69.34|66.19|71.24|
>
> **Table 2. Regression (MAE, ↓)**
> |Task|GraphSAGE|RelGNN w/ GraphSAGE|GAT|RelGNN|
> |-|-|-|-|-|
> |User-ltv|14.313|14.240|16.626|14.230|
> |Item-ltv|50.053|48.282|58.902|48.767|
> |Ad-ctr|0.041|0.037|0.043|0.037|
> |User-attendance|0.258|0.238|0.263|0.238|
> |Driver-position|4.022|3.792|4.268|3.798|
> |Item-sales|0.056|0.053|0.079|0.054|
> |Post-votes|0.065|0.065|0.068|0.065|
> |Study-adverse|44.473|45.531|46.026|44.461|
> |Site-success|0.400|0.354|0.393|0.301|
>
> **Table 3. Recommendation (MAP, ↑)**
> |Task|GraphSAGE|RelGNN w/ GraphSAGE|GAT|RelGNN|
> |-|-|-|-|-|
> |user-item-purchase|0.74|0.79|0.44|0.77|
> |user-item-rate|0.87|0.91|0.78|0.92|
> |user-item-review|0.47|0.57|0.29|0.52|
> |user-ad-visit|3.66|3.95|1.99|3.94|
> |user-item-purchase|2.81|2.82|1.88|2.81|
> |user-post-comment|12.72|13.9|11.97|14|
> |post-post-related|10.83|11.79|10.71|11.66|
> |condition-sponsor-run|11.36|11.62|10.43|11.55|
> |site-sponsor-run|19.00|19.17|17.90|19.14|
>
> > **Regarding Format (C3)**
>
> We thank the reviewer for pointing this out and will revise the paper.
>
> > **Regarding Complexity in Large DBs (Q2)**
>
> The reviewer asks if the complexity of computing atomic routes increases with larger DBs. The answer is no. Atomic routes are computed at the schema graph level—where nodes represent table types, which typically number in the tens—even for very large databases. This design choice makes our method agnostic to the number of individual entities (which can be in the millions), allowing RelGNN to scale efficiently as databases grow while the number of table types remains relatively stable. We will clarify this in the revised version.
>
> > **Regarding Handling Tables with Many Foreign Keys (Q3)**
>
> The reviewer is concerned about complexity when handling tables with many foreign keys. We employ a sampling method (following the standard RDL implementation), so only a subset of nodes around the seed node is sampled. Consequently, we only process foreign keys within the sampled subset. The sample size can be controlled to ensure that the number of connections remains manageable. We will explain this in the revised version.
>
> In summary, we provide potential solution for self-loop tables, discuss integration with advanced techniques, provide the ablation study that confirms the effectiveness of atomic routes and clarify how RelGNN is capable of handling large and complex DBs efficiently. We hope these clarifications address the reviewer’s concerns and will update the manuscript when permitted.

---

> > ### Comment · Reviewer_935g · 2025-04-02
> >
> > Many thanks for the rebuttal and extra experiments. Yet I am still concerned about the changes going to happen in the paper. For example, Regarding Handling Self-Loop Tables (W1, C2, Q1), and Regarding Discussion of Integrating Other Components (W2). Although the authors mentioned will include a discussion in the paper, the discussion details are unclear in this rebuttal.

---

> > > ### Author Response · Authors · 2025-04-04
> > >
> > > We thank the reviewer for the prompt response and for acknowledging the additional experiments. Since we cannot revise the paper at this stage, we provide the exact content we plan to include in the revision to address the concerns.
> > >
> > > > **Handling Self-Loop Tables** (W1, C2, Q1)
> > >
> > > We'll include:
> > >
> > > Although self-loop tables are rare in relational databases (only 1 out of 7 datasets in RelBench), improving model handling of such cases is worthwhile. The key to addressing this challenge is enhancing the model's ability to distinguish messages coming from the same table type (for self-loop tables) versus those from different types (for non-self-loop tables). To address this, we propose incorporating relative positional encoding (RPE) over the schema graph (where nodes are table types and edges are primary-foreign key relations). RPE helps the model identify whether two tables are of the same type and also offers additional benefits. Since the schema graph defines the database structure at a macro level, it can provide global relational context that complements the local structural information captured by GNNs on the data graph. Moreover, schema graphs are typically small (with only tens of nodes even in large databases) and static, so the RPE can be precomputed once with negligible overhead and reused throughout training and inference.
> > >
> > > To evaluate, we implemented an RPE method based on [1]: eigen-decompose the Laplacian of the schema graph $L=V\text{diag}(\lambda)V^{\top}$, apply $m$ element-wise MLPs $\phi_k(\cdot)$ to obtain $Q[:,:,k]=V\text{diag}(\phi_k(\lambda))V^{\top}$, for $k\in[m]$, resulting in a tensor $Q\in\mathbb{R}^{N\times N\times m}$, and project via an MLP $\rho$ to obtain $RPE=\rho(Q)\in\mathbb{R}^{N\times N\times d}$, where $N$ is the number of node types of $d$ is the dimension of node embeddings. For a message from node type $i$ to $j$, we update the original message $m$ as $m' = m + \alpha \cdot RPE[i, j]$ with learnable $\alpha$. This resulted in a 2% improvement on both classification tasks in rel-stack, which contains self-loops.
> > >
> > > > **Integrating Other Components** (W2)
> > >
> > > We'll include:
> > >
> > > We focus on improving the core design of message passing, but the RelGNN framework is flexible and accommodates more advanced components.
> > >
> > > **Advanced GNN Aggregators**. RelGNN allows easy integration of different GNN aggregators by re-instantiating the AGGR operation in Eq. 5. For example, PNA [2] improves expressiveness by combining multiple aggregators and degree-scalers. It can be integrated into RelGNN by instatiating Eq. 5 with
> > > $AGGR(h_{dst}^{(l)},${{$h_{fuse}^{(l)}$}}$)=W_{\text{proj}}h_{dst}^{(l)}+\sum_{fuse\in \mathcal{N}(dst)} M(h_{fuse}^{(l)})$
> > >  where $M(\cdot)$ is the PNA operator. Model-agnostic aggregators can also be integrated on top of RelGNN. For instance, Jumping Knowledge Network [3] improves performance by adaptively combining representations from multiple layers. This can be implemented by collecting outputs from each RelGNN layer and passing them to PyG’s `JumpingKnowledge` module, with the result used as input to the final prediction layer.
> > >
> > > **Effective Time Encodings**. Many RDL tasks involve temporal prediction, and various temporal encodings can be incorporated as additional node features for RelGNN. For example, Time2Vec [4] maps timestamps $t$ to $\mathbb{R}^d$ via $\text{Time2Vec}(t)[i]=sin(\omega_it+\phi_i)$ with learnable parameters  $\omega_i$ and $\phi_i$. GraphMixer [4] introduces a fixed time encoding $t \rightarrow \cos(t \omega)$, where $\omega=${$\alpha^{-\frac{i-1}{\beta}}$}$_{i=1}^d$ and the choice of $\alpha,\beta$ depends on the dataset's time range. These techniques are compatible with RelGNN and introduce minimal overhead.
> > >
> > > > **Other Changes** (C1, C3, Q2, Q3)
> > >
> > > We will
> > >
> > > - include ablation results in Section 4 (C1)
> > > - fix formatting issues (C3)
> > > - clarify how RelGNN scales to large DBs (Q2) and handles many foreign keys (Q3) in Section 3.2, elaborating on atomic routes and sampling strategy as described in the original rebuttal.
> > >
> > > We hope these specific additions address the reviewer’s concerns and demonstrate our commitment to improving the final version.
> > >
> > >
> > > References:
> > >
> > > [1] Huang et al. "On the stability of expressive positional encodings for graphs." ICLR, 2024.
> > >
> > > [2] Corso et al. "Principal neighbourhood aggregation for graph nets." NeurIPS, 2020.
> > >
> > > [3] Xu et al. "Representation learning on graphs with jumping knowledge networks." ICML, 2018.
> > >
> > > [4] Kazemi et al. "Time2vec: Learning a vector representation of time." arXiv:1907.05321.
> > >
> > > [5] Cong et al. "Do we really need complicated model architectures for temporal networks?." ICLR, 2023.

---

### Official Review · Reviewer_pL7w · 2025-03-13

**Overall Recommendation:** 1

**Summary:**

The manuscript proposes RelGNN, a graph neural network framework tailored for relational deep learning (RDL), enabling predictive modeling on relational databases. RelGNN introduces atomic routes, which capture high-order tripartite structures to facilitate direct single-hop interactions between heterogeneous nodes. By designing a composite message passing mechanism based on atomic routes, RelGNN completes the two-step information exchange in a single step with an attention mechanism. RelGNN is empirically evaluated on 30 tasks from RelBench [1] which consists of entity classification, entity regression, and recommendation.

[1] Fey et al., “Relational Deep Learning - Graph Representation Learning on Relational Databases”, ICML Position Paper, 2024.



## Update after rebuttal
The paper could benefit from including additional evidence to support the main claim and existing heterogeneous GNNs with relevant adjustments in the experiments.

**Claims And Evidence:**

In Section 3.1, the authors claim that standard heterogeneous GNNs entangle irrelevant information when propagating messages through intermediate nodes with two or more foreign keys. However, there is insufficient evidence to support the claim that messages propagated from one-hop or two-hop neighboring nodes of an intermediate node contribute as irrelevant noise. For instance, the authors argue that the “constructors” node introduces noise during the message-passing process to the “standings” node. However, even in a two-hop scheme, the “constructors” node does not contribute to the message-passing process to the “standings” node. Furthermore, in relational databases, the assumption that the information from tables connected via primary-foreign key relationships to a table corresponding to an intermediate node cannot be deemed irrelevant contradicts the rationale for using GNN-based models. Therefore, a clear theoretical justification for this claim is necessary.

**Essential References Not Discussed:**

References and discussions of the various baseline models mentioned in the “Experimental Designs or Analyses*” section are necessary [2,3,4,5,6].

**Experimental Designs Or Analyses:**

The baseline models for evaluating the proposed RelGNN are highly limited. The authors primarily use only the heterogeneous GraphSAGE from the original RelBench [1] as the baseline. However, it is necessary to compare RelGNN with other heterogeneous GNNs that employ different backbone GNN architectures, such as GAT [2] and GIN [3], rather than relying solely on GraphSAGE. Additionally, other heterogeneous GNNs including models that use meta-paths should be included as baselines [4,5,6]. Furthermore, despite the use of only a limited baseline, the performance improvement of RelGNN is not significant.

[2] Veličković et al., “Graph Attention Networks”, ICLR, 2018.\
[3] Xu et al., “How Powerful are Graph Neural Networks?”, ICLR, 2019.\
[4] Fu et al., “MAGNN: Metapath Aggregated Graph Neural Network for Heterogeneous Graph Embedding”, WWW, 2020.\
[5] Hu et al., “Leveraging Meta-path based Context for Top-N Recommendation with A Neural Co-Attention Model”, KDD, 2018.\
[6] Tang et al., “HiGPT: Heterogeneous Graph Language Model”, KDD, 2024.

**Methods And Evaluation Criteria:**

RelGNN is evaluated on the recently proposed RelBench [1]. There is no new benchmark dataset or evaluation criteria.

**Other Comments Or Suggestions:**

- In line 373 of the main text, the phrase starting with “tianlangNote to self” in the right column may compromise anonymity.
- The terms “Relational Entity Graph” and “Relational Data Graph” appear to refer to the same concept, yet they are used inconsistently throughout the manuscript.

**Other Strengths And Weaknesses:**

- The authors provide reasonable explanations for the experimental results of the proposed model, particularly when its performance is comparable to or worse than the baseline models. However, most limitations have been left as future work without being addressed. Additionally, further discussion is needed after incorporating other baseline models.
- There is a lack of justification for key model components, such as the attention mechanism and FUSE operation. In particular, without an ablation study on the attention mechanism, it is unclear whether the model’s performance improvements are due to the introduction of atomic routes or the attention mechanism itself.
- The manuscript lacks a clear explanation of the model’s training methodology. While it can be inferred that the training table was used following previous work [1], this is not explicitly stated, leaving the details of the training and inference process ambiguous. Additionally, the meaning of the gray nodes in Figure 2(c) is not clearly explained, making it difficult to interpret their role.
- The code/data for the experiments is not provided, and the detailed descriptions of the experimental setup (e.g., number of layers) are insufficient, making it difficult to ensure reproducibility.

**Questions For Authors:**

None

**Relation To Broader Scientific Literature:**

In this manuscript, RelGNN is proposed for predictive tasks on relational databases. However, the rationale behind introducing its main idea, the atomic route, is not clearly justified. Additionally, the limited number and variety of baseline models make it difficult to properly assess the significance of RelGNN.

**Theoretical Claims:**

There is no formal proof or theoretical justification for the claims made in this manuscript.

---

> ### Author Rebuttal · Authors · 2025-04-01
>
> We thank the reviewer for the feedback and address each concern as follows:
>
> > Regarding Justification of Claims
>
> The message passing process through an intermediate node introduces an imbalance: the source node’s signal is aggregated twice, while signals from intermediate nodes are only aggregated once. In RDL, the source nodes typically contain the most critical information for prediction, so preserving a clean, undiluted source signal is essential. Standard heterogeneous architectures attempt to address this issue by employing skip connections and additional components such as MLPs, which must implicitly learn to disentangle the clean source signal from the noise introduced during multiple aggregations. In contrast, our approach explicitly leverages atomic routes and composite message passing to maintain a direct, efficient exchange of information between source, intermediate, and destination nodes. This targeted mechanism minimizes the dilution of the source signal without introducing additional parameters, ensuring that the most critical information propagates effectively through the network.
>
> > Regarding Additional Baselines
>
> The reviewer asks for additional baselines. Our original choice of baselines was intended to maintain consistency with prior work [1]. In response to the reviewer’s suggestion, we conducted experiments using GAT [2] and GIN [3] as backbone architectures. As presented in Table 1, the performance remains largely unchanged across different backbone GNN architectures.
>
> Regarding meta-path-based GNNs, these methods require substantial manual effort and domain expertise to select appropriate meta-paths, making them less directly applicable to RelBench. In contrast, our method is fully automated and eliminates the need for such human intervention. We have already discussed meta-path approaches and cited [4,5] in the original manuscript, and we will add the citation [6] in the revised version.
>
> > Regarding Limitations
>
> The reviewer notes that some limitations are left as future work. We clarify that addressing most of these limitations is orthogonal to our main contribution. RDL is an emerging field, and the pipeline described in [1] opens up research opportunities across multiple dimensions. Our work focuses on the foundational aspect of improving message passing design, while complementary aspects—such as refined prediction heads and improved recommendation frameworks, noted as limitations of the current pipeline—are recognized as independent. By keeping these components constant, we isolate the effect of our proposed message passing method, ensuring a fair comparison. We highlight these limitations to encourage further advancements in the field.
>
> > Regarding Ablation on Model Components
>
> We appreciate the reviewer’s suggestion for the ablation study.  The FUSE operation of RelGNN is instantiated in the same way as GraphSAGE in our original manuscript. For the ablation study, we further instantiate the AGGR operation with GraphSAGE instead of attention so that the only difference from the baseline is the inclusion of atomic routes. As shown in Table 1-3, the performance gap clearly demonstrates the effectiveness of atomic routes. The attention mechanism itself does not contribute significantly to performance gains (RelGNN w/ attn vs RelGNN wo/ attn; GraphSAGE baseline vs GAT baseline). We will include further details of this ablation study in the revised version.
>
> Table 1. Classification (ROC-AUC, ↑)
>
> Table 2. Regression (MAE, ↓)
>
> Table 3. Recommendation (MAP, ↑)
>
> > Regarding Code and Other Details
>
> We've now made our source code and model checkpoints available at https://anonymous.4open.science/r/RelGNN for reproducibility. The training setup and experimental details are maintained consistent with [1] to ensure a fair comparison. In the revised manuscript, we will add a section detailing these setup.
>
> The gray nodes in Figure 2(c) denote training tables, derived in the same manner as in [1]. “Relational Entity Graph” specifically refers to the graph built from relational tables that encompasses all entities, while “Relational Data Graph” is used more generally to refer to any graph constructed from a relational database (e.g., a subgraph sampled from the Relational Entity Graph). We will clarify these points in the revised manuscript.
>
>
> We hope these clarifications address the reviewer’s concerns and will update the manuscript when permitted.

---

### Decision · Program_Chairs · 2025-05-01

**Decision:**

Accept (poster)

**Comment:**

The papers presents RELGNN, a novel graph neural network framework that improves predictive modeling of relational databases by utilizing atomic routes and composite message passing to capture relational structures effectively. Doing so achieves up to 25% improvement in accuracy over existing models across various real-world tasks. One reviewer points out some downsides such as a potentially unproven claim that standard heterogeneous GNNs entangle irrelevant information,  weak selection of baselines, as the paper mainly uses heterogeneous GraphSAGE as a comparison and a missing ablation study. These downsides have been largely addressed in the rebuttal. While indeed more existing heterogenuous GNNs could be included, the updated results (including ablation studies) as well as the overall idea underlying RELGNN, seem to be interesting, as pointed out by the other reviewers. The other reviewers also stress the novelty of the approach as well as relatively strong results on a large set of real tasks from RELBENCH. Therefore, I overall recommend to accept the paper, overruling one of the reviewers. I agree with them.